# The Intuition of Punishment: A Study of Fairness Preferences and Cognitive Ability

**Markus Seier**

Department of Economics and Business Economics, Aarhus University, 8210 Aarhus V, Denmark;
msp1996@live.dk

**Abstract:** Can differences in cognitive reflection explain other-regarding behavior? To test this, I use the three-item Cognitive Reflection Task to classify individuals as *intuitive* or *reflective* and correlate this measure with choices in three games that each subject participates in. The main sample consists of 236 individuals who completed the *dictator game, ultimatum game* and a *third-party punishment task.* Subjects afterwards completed the three-item Cognitive Reflection Test. Results showed that *intuitive* individuals acted more prosocially in all social dilemma tasks. These individuals were more likely to serve as a norm enforcer and *third-party punish* a selfish act in the *dictator game. Reflective* individuals were found more likely to act consistently in a self-interested manner across the three games.

**Keywords:** social preferences; third-party punishment; cognitive reflection ability; intuition; reflection; dictator game; ultimatum game

---

## 1. Introduction

Human societies depend on their members acting cooperatively. Social sanctioning is crucial for the maintenance of cooperative behavior when there exist material incentives to deviate from collectively desirable behavior, such as benefiting from a public good without bearing the cost of contributing. Sanctioning behavior can be explained by strong reciprocity, which is defined by a willingness to sacrifice resources to reward cooperative actions and to punish hostile actions even when this is costly and provides neither present nor future material rewards for the reciprocator [1,2]. Thus, individuals acting as norm enforcers enable cooperative behavior because of an understanding and expectation that a deviation will be sanctioned [3]. Social dilemma experiments reveal a great deal of strong reciprocity. For example, in [4], the majority of subjects were willing to engage in *third-party punishment*. That is, they punished a hostile action even though it did not affect their personal earnings.

Is sanctioning a norm violation an *intuitive* response, or does it take *deliberation* to sacrifice resources? To the best of my knowledge this question has not been investigated in the context of *third-party punishment*, where there is no indirect benefit from sanctioning through reputation-building or long-term material incentives from changing the behavior of people one interacts with in the future.

More generally, is cooperative behavior driven by an *intuitive* response or due to *deliberation*? Whether individuals rely on *intuition* or *reflection* in social dilemma experiments has been shown to generate differences in behavior. Applying cognitive reflection tests [5,6], subjects relying on *intuition* in decision-making are found to act more prosocially [7–11].

I contribute to this literature by examining whether behavior is consistent across three games and whether sanctioning the violation of a norm is an *intuitive* action. Applying a *third-party punishment task*, subjects are given the opportunity to, at a personal cost, sanction another subject who kept the entire endowment to herself in the *dictator game.*

Studying subjects' response time has as well been applied to access whether individuals rely on *intuition* in decision-making. Results in these studies are, however, not conclusive about whether a

faster response time indicate more prosocial [12] or more egoistic [13] behavior. Identifying whether a choice is *intuitive* or *deliberate* from response time suffers from endogeneity issues as various cognitive processes contribute to response time. When controlling for strength-of-preference, there is no evidence that one type of choice is systematically faster than the other [14].

According to the Social Heuristic Hypothesis, *intuitive* individuals behave more prosocially in the lab because they internalize generally beneficial behavior from daily life that favors cooperative and fair behavior [15]. In light of this, the employed experimental design in this study investigates how strong these internalized fairness preferences are.

The purpose of this study is twofold. By having subjects complete the *dictator game*, the *ultimatum game* (both in the role of *proposer* and *recipient*) and finally the *third-party punishment task* the purpose is first to see if subjects display consistent behavior across games in line with the hypothesis that the "fair" outcome drives *instinctive* choices but that it takes *deliberation* to act selfishly. Secondly, this experiment investigates for the first time if the *instinctive* action is to engage in *third-party punishment* toward a *dictator* who kept the entire endowment to herself in the *dictator game*. The subjects' tendency to rely on *intuition* in decision-making is assessed by Frederick's three-item Cognitive Reflection Test (CRT) [5].

The sample consists of 295 students at Aarhus University, collected during spring 2019.

The results of this study confirmed, first of all, previous findings that *reflective* subjects act more selfishly and in accordance with the economic prediction in the *dictator-* and *ultimatum games*. They transferred less in *dictator game*, they offered less as the *proposer* in the *ultimatum game*, and they were more likely to accept a low offer as *recipient*. Secondly, the experiment extended previous findings to *third-party punishment* by showing that the *intuitive* action was to sanction a norm-violator. Subjects relying on *intuition* in decision-making were found more likely to sacrifice resources to sanction a *dictator* who kept the entire endowment to herself. Taken together, the results of this experiment provide evidence that the *intuitive* action is to engage in "fair" behavior, or to sanction those not complying with the social norm of fair behavior.

In the following Section 2, I present the experimental design. The hypotheses are presented in Section 3. Section 4 presents the results of the experiment. Section 5 provides a general discussion of the findings. Section 6 discusses the limitations of this study. Section 7 concludes.

## 2. Experimental Design

### 2.1. Procedures

Subjects were recruited during four lectures in Psychology, Political Science, and Economics at Aarhus University. Three of these four lectures were for second semester students. The students were orally encouraged to participate during the break of the course and a link to the survey was distributed online.

Subjects were incentivized through a lottery scheme. In total, seven subjects were paid on average DKK 50 (≈$7.5) for completing the experiment and, in pairs, paid according to their choices in the task, for which they were drawn at random to receive payment. For each the *dictator game*, *ultimatum game*, and *third-party punishment task*, two subjects received payment. One subject was drawn to get paid for completing the CRT. For each correct answer on the CRT, one ticket was added to the bowl from where a subject was drawn. The subject received DKK 100 (≈$15) for completing the CRT.

### 2.2. Experimental Design

Subjects completed four social dilemma tasks: The *dictator game*, the *ultimatum game* with role uncertainty (i.e., subjects made choices in the role of both the *proposer* and the *recipient*) and decided whether to engage in *third-party punishment* by choosing if and how much to sacrifice to sanction a *dictator*, who kept the entire endowment to herself in the *dictator game*. After completing the four social dilemma tasks, subjects continued to the second part of the experiment to complete the three-item CRT.

Lastly, subjects were to state their gender, line of study and their email address in order to potentially get paid for participating in the experiment.

In the following, I will present each social dilemma task as well as the three-item CRT. The experimental instructions are reproduced in Appendix A.

### 2.2.1. Dictator Game

The first task was a standard *dictator game*. The subject acting in the role of the *dictator* was endowed with DKK 100 and had to decide on how much (in increments of DKK 10) to transfer to another subject acting as the *receiver*, with whom she was randomly matched. The *receiver* had no decision to make.

### 2.2.2. Ultimatum Game

For the second and third task, subjects were to make a decision first as *proposer* and later as *recipient* in the *ultimatum game*. The *proposer* is endowed with DKK 100 and chooses how much to offer (in increments of DKK 10) the *recipient*. The *recipient* indicates the minimum amount (acceptance threshold), she is willing to accept (in increments of DKK 10). If the offer is accepted, the proposed allocation is realized, and if the offer is rejected, both the *proposer* and the *recipient* receive nothing.

The strategy method [16] is employed to the *recipient's* decision because the sampling procedure allowed players to enter their choices at different time points. Even though applying the strategy method was necessary in this case, it is useful in the *ultimatum game*, since most offers are close to equal splits which means that there are few rejections, and thereby the actually relevant choices provide little information regarding the willingness to accept or reject low offers [17].

### 2.2.3. Third-Party Punishment Task

The fourth and final social dilemma task added a *third-party punishment* option to the *dictator game*. The subject is informed that she has been randomly matched to a pair of other subjects from the *dictator game*. One of the other subjects was assigned to the role of the *dictator* and chose to keep the entire endowment to herself[1]. The subject, who must decide on how much (if at all) to punish the *dictator* is endowed with DKK 50. For each DKK 1, the *third-party punisher* sacrifices, the *dictator* suffers a reduction in earnings of DKK 5. The *third-party punisher* must decide on how much to sacrifice between DKK 0 and DKK 20. By sacrificing DKK 20 of her own endowment, the *third-party punisher* can reduce the earnings of the *dictator* to DKK 0.

### 2.2.4. Three-Item Cognitive Reflection Test

After having completed the above-mentioned tasks, the subjects proceed to the three-item CRT [5]. The three-item CRT can be found in Appendix B.

The test is used to detect an individual's proclivity for applying two systems of decision-making: System 1 and System 2 processes [19]. System 1 is the intuitive "part" of the brain that relies on heuristics and automaticity. It possesses no computational capacity and is characterized as unconscious. It is fast, automatic and requires no effort. System 2 is the more analytical and rational system. It is deliberate and activated when facing complex calculations, different choices and requires the individual to be focused [20]. The performance on CRT indicates whether an individual is able to overcome the desire to go with the intuitive (incorrect) answer, reflect further upon the question and reach the, when explained to, relatively easy correct answer. For example the first question of the CRT: *A bat and*

---

[1] The experimental design applied actual matching on the subset of subjects who gave DKK 0 in the *dictator game*. Ex ante it could be expected that at least one subject would do so, based on previous *dictator game* experiments (In a meta study [18] found that 36.11% of all participants chose to give nothing).

*a ball cost $1.10. The bat costs $1.00 more than the ball. How much does the ball cost? ___ cents. Intuitive Answer: 10 / Correct Answer: 5.*

Based on the answers to the CRT I divide subjects into three groups using the categorization used by [21]: Subjects who answered correctly two or more items on the CRT are categorized as *reflective*. Those opting for the intuitive, but wrong answer at least in two of the three items are *intuitive*. The subjects who are not categorized as either *reflective* or *intuitive*, form the residual group. For precise details of the categorization, see Appendix C.

## 3. Hypotheses

Looking to replicate previous findings of fair behavior by individuals relying on *intuition* in decision-making and that it takes *reflection* to pursue a self-interested objective gives three hypotheses in the *dictator-* and *ultimatum game* decisions.

**Hypothesis 1.** *Reflective individuals transfer less in the dictator game compared to intuitive individuals.*

**Hypothesis 2.** *Reflective individuals offer less as proposer in the ultimatum game compared to intuitive individuals.*

**Hypothesis 3.** *Reflective individuals require a smaller share to accept the offer as ultimatum game recipient compared to intuitive individuals.*

Including both the *proposer* decision in the *ultimatum game* and the transfer decision in the *dictator game*, it is possible to detect whether strategic considerations drive the *ultimatum game* offer. In the *dictator game,* such strategic considerations are absent, because it is a pure decision problem without strategic interaction. Expecting the *intuitive* action to be fair and *reflection* to lead to rational, self-interested decisions generates two hypotheses for *proposer* and *dictator* behavior.

**Hypothesis 4a.** *Reflective individuals offer more in the ultimatum game relative to their transfer in the dictator game.*

**Hypothesis 4b.** *Intuitive individuals do not offer more in the ultimatum game relative to their transfer in the dictator game.*

A main contribution of this study is the investigation of whether the *intuitive* action is to sanction those who violated the norm of fair behavior.

**Hypothesis 5.** *Intuitive individuals exhibit a greater willingness to punish a selfish dictator than reflective individuals.*

The other contribution to the existing literature is that this study investigates the behavior across four social dilemma decisions.

**Hypothesis 6.** *Reflective individuals act consistently more rational and self-interested in the four social dilemma decisions compared to intuitive individuals.*

## 4. Results

A total of 295 subjects completed the study. The main sample consists of 236 observations, for which all variables of interest are available. Of the 236 subjects in the main sample, 124 (52.5%) were male subjects (one subject did not state gender). 214 of the subjects were students at the faculty of Business and Social Sciences at Aarhus University, which leaves a minority from other faculties. This is not surprising, because the courses where the study was advertised are available in the faculty of Business and Social Sciences.

In each task, a few subjects chose the opposite extreme of strict self-interest (transferring DKK 100 in the *dictator game* and offering DKK 100 in the *ultimatum game* and accepting no less than DKK 100 in the *ultimatum game*). These "outliers" are included in the analysis. Excluding them does not alter the findings.

### 4.1. Cognitive Reflection Test Results

On average, the subjects answered 2.1 of the items on the CRT correctly. Of the 236 subjects, 48.7% answered all three items correctly, 24.2% answered two correctly, 14.4% answered one correctly and 12.7% did not answer any of the three items correctly. 9% of the subjects opted for the *intuitive* incorrect answer in all three items, 23.3% chose the *intuitive* answer in at least two items and 45.8% chose the *intuitive* incorrect answer at least once.

The *reflective* group consists of 172 subjects. The *intuitive* group consists of 56 subjects. The residual group consists of 8 subjects. As the residual group consists only of 8 subjects, these are grouped with the *intuitive* subjects throughout the statistical analysis. Therefore, the analyses mainly compares those *reflective* to those not *reflective*. The *non-reflective* group consists therefore of 64 subjects. Excluding the residual group, and thereby comparing the *reflective* to the *intuitive* subjects, does not change conclusions. (See Appendix D (Tables A1–A6, Figures A1–A5) for a summary of the findings excluding the residual group).

Men performed better in the CRT by answering an average of 2.3 items correctly compared to women with an average of 1.84 correct answers. This difference is statistically significant ($p = 0.003$, MWU[2]). The distribution of the answers can be found in Appendix E (Tables A7–A11).

In the following subsections, I will present the results for each of the tasks in the experiment. A graphical representation of the frequency of decisions consistent with rational, self-interested behavior by *non-reflective (reflective)* individuals can found in Figure 1. A more detailed presentation of decisions in each task can be found in Appendix F (Tables A12–A16, Figures A6–A9).

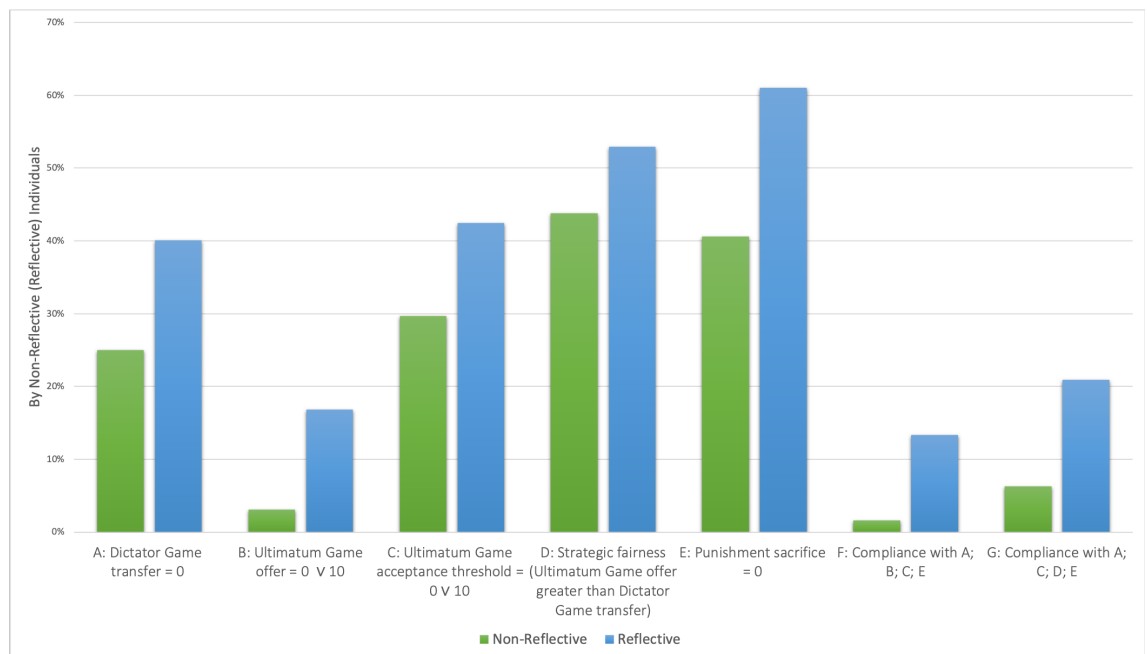

**Figure 1.** Frequency of Decision by Non-Reflective (Reflective) Individuals.

---

[2]    Mann-Whitney-U: Note that the MWU is a test of differences in distribution.

*4.2. Dictator Behavior*

*Result 1: Reflective subjects transfer less in the dictator game than intuitive subjects.*

*Reflective* subjects transfer on average less than those not *reflective* (average transfer of DKK 28.2 and DKK 36.4, respectively). This difference is statistically significant at the 5% significance level ($p = 0.03$, MWU).

The average amount transferred to the *recipient* in the *dictator game* was DKK 30.4. The modal transfer was DKK 50, which 44.5% of the subjects chose, whereas 36% of the subjects chose to keep the entire endowment to themselves.

Transferring 0 DKK to the *receiver* and thus comply with the prediction from standard economic theory is more common for the *reflective* subjects (40.1% chose this versus 25% of the *non-reflective*). This difference is statistically significant at the 5% level ($p = 0.032$, $\chi^2 - test$). However, a part of the difference can be contributed to gender: Males are found significantly more likely to transfer DKK 0 to the *receiver* in the *dictator game* Thus, it appears that acting selfish in the *dictator game* is independent of being *reflective* when controlling for gender. Gender seems to be the significant factor that predicts behavioral differences (see Table 1).

*4.3. Proposer Behavior in the Ultimatum Game*

*Result 2: Reflective subjects offer less in the ultimatum game than intuitive subjects.*

*Reflective* subjects offer on average less than those not *reflective* (average offer of DKK 40.9 and DKK 50.5, respectively). This difference is statistically significant ($p = 0.0001$, MWU).

The average offer in the *ultimatum game* was DKK 43.5. The most frequently offered amount was DKK 50, which 68.6% of the subjects chose.

Of the *reflective* subjects, 15.7% offered DKK 10. Only one subject (1.8%) from the *intuitive* group offered DKK 10.

Distinguishing whether the *recipient* accepts or rejects an offer when indifferent, both offers of DKK 0 and DKK 10 can be considered consistent with rational and strictly self-interested behavior. 16.9% of the *reflective* subjects chose either of these offers as opposed to 3.1% of the *non-reflective*. This difference is statistically significant ($p = 0.005$, $\chi^2 - test$).

When controlling for gender, *reflective* subjects are estimated to be 12.6%-points more likely than *non-reflective* subjects to offer DKK 0 or DKK 10 in the *ultimatum game*. *Reflective* subjects are predicted to choose such an offer with a probability of 16.2% as opposed to a predicted probability of 3.6% for those *non-reflective* (see Table 1).

*4.4. Recipient Behavior in the Ultimatum Game*

*Result 3: Reflective subjects are willing to accept lower offers in the ultimatum game than intuitive subjects.*

*Reflective* subjects have on average a lower acceptance threshold relative to those not *reflective* (average threshold of DKK 27.8 and DKK 33.9, respectively). This difference is statistically significant at the 5% significance level ($p = 0.032$, MWU).

The average acceptance threshold was DKK 29.45. The modal acceptance threshold was DKK 10 and was chosen by 32.2% of the subjects whereas DKK 50 (requiring an equal split) was chosen by 29.7% of the subjects.

For the *reflective* subjects, the modal acceptance threshold was DKK 10, which was chosen by 36.6% in this category as opposed to 21.4% in the *intuitive* category. The modal acceptance threshold for the *intuitive* subjects was DKK 50, which was chosen by 37.5% in this category as opposed to 25% in the *reflective* category.

Both an acceptance threshold of DKK 0 or DKK 10 can be considered the rational, self-interested choice. 42.4% of the *reflective* subjects chose one of these thresholds as opposed to 29.7% of the *non-reflective* subjects. This difference is statistically significant at the 10% significance level ($p = 0.074$, $\chi^2 - test$). When controlling for gender, *reflective* subjects are estimated to be 11.4%-points more likely,

compared to *non-reflective* subjects, to choose an acceptance threshold of DKK 0 or DKK 10 as *recipient* in the *ultimatum game*. *Reflective* subjects are predicted to choose such an acceptance threshold with a probability of 42.2% as opposed to a predicted probability of 30.8% for those *non-reflective* (see Table 1).

### 4.5. Dictator/Proposer Comparison

*Result 4: Both reflective and intuitive subjects increase their offer in the ultimatum games relative to their transfer in the dictator game.*

Across all subjects, the average transfer in the *dictator game* was DKK 30.4 and the average offer in the *ultimatum game* was DKK 43.5. Applying a Wilcoxon Sign Rank test, these means are significantly different ($p < 0.001$). Applying the test when distinguishing between *reflective* and *intuitive* subjects yields the same conclusion ($p's < 0.001$). Thus, both the *reflective* and *intuitive* subjects increase their offer in the *ultimatum game* relative to their transfer in the *dictator game*.

More than half of the subjects (50.4%) chose to increase their offer in the *ultimatum game* compared to their transfer in the *dictator game*—exhibiting strategic fairness. 52.9% of the *reflective* and 43.8% of the non-*reflective* subjects opted for this decision. This difference is not statistically significant ($p > 0.21$, $\chi^2 - test$).

When controlling for gender, *reflective* subjects are estimated to be 4%-points more likely to exhibit strategic fairness than *non-reflective* subjects. However, the effect is not statistically significant. *Reflective* subjects are predicted to exhibit strategic fairness with a probability of 51.3% as opposed to a predicted probability of 47.3% for those *non-reflective* (see Table 1).

### 4.6. Third-Party Punishment Behavior

*Result 5: Intuitive subjects are more likely to punish a selfish dictator than reflective subjects.*

Of the 236 subjects, 105 chose to punish the *dictator*, who kept the entire endowment to herself. The average amount sacrificed was DKK 4.8 which implies that a selfish *dictator*, on average, had her income reduced by DKK 24. The modal amount sacrificed was DKK 0, which 55.5% of the subjects chose. 10.2% of the subjects chose to reduce the earnings of the selfish *dictator* to DKK 0 by sacrificing DKK 20 of their endowment. 15.3% of the subjects chose to reduce the *dictator's* earnings by DKK 50 leaving the *dictator* with half of her initial endowment.

57.1% of the *intuitive* subjects chose to punish as opposed to 39% of the *reflective* subjects. This difference is statistically significant ($p = 0.017$, $\chi^2 - test$). The *reflective* subjects sacrificed, on average, DKK 3.97 as opposed to DKK 6.69 sacrificed by *intuitive* subjects. This difference is statistically significant ($p < 0.01$, MWU). Comparing the *reflective* subjects to those not *reflective* yields the same conclusion.

Considering only the subjects who opted for the opportunity to *punish* the selfish *dictator*, the *intuitive* subjects sacrificed, on average, DKK 11.7 as opposed to DKK 10.2 by the *reflective* subjects. This difference is not statistically significant ($p > 0.32$, MWU).

When controlling for gender, *reflective* subjects are estimated to be 20.1%-points more likely to not punish the dictator than *non-reflective* subjects. *Reflective* subjects are predicted to not engage in *third-party punishment* with a probability of 61.2% as opposed to a predicted probability of 41.1% for those *non-reflective* (see Table 1).

### 4.7. Consistency in Choices

*Result 6: Reflective subjects are more likely to act consistently and in line with rational, self-interested behavior across all social dilemma tasks compared to intuitive subjects.*

A rather clear prediction for rational, self-interested behavior exists for the *dictator game*, *recipient's* acceptance threshold in the *ultimatum game*, and the *third-party punishment task*. However, the decision as *proposer* in the *ultimatum game* is rather difficult to classify as expectations for the decision of the *recipient* matter. Thus, any offer can be considered rational, self-interested if that is the lowest amount the *proposer* expects to be accepted.

Due to the ambiguity in what constitutes rational, self-interested behavior in the *ultimatum game* *proposer* decision, I will consider offering DKK 0 or DKK 10 and strategic fairness separately.

First I consider whether *reflective* subjects are more likely to transfer DKK 0 in *dictator game*, offer DKK 0 or DKK 10 as *proposer* in the *ultimatum game*, acceptance threshold of DKK 0 or DKK 10 as *recipient* in the *ultimatum game* and not opting for the *punishment* opportunity in the *third-party punishment task*.

13.4% of the *reflective* subjects complied with the above-mentioned as opposed to 1.6% of those not *reflective*. This difference is statistically significant ($p = 0.008$, $\chi^2 - test$). When controlling for gender, *reflective* individuals are predicted to be 10.8%-points more likely than *non-reflective* subjects to choose as described in these tasks. *Reflective* subjects are predicted to choose as described with a probability of 12.7% as opposed to a predicted probability of 1.9% for those *non-reflective* (see Table 1).

A rational, self-interested individual could, as *proposer* in the *ultimatum game*, offer any share to the *recipient* if this is what the *proposer* believes to be the lowest amount to be accepted. However, in the *dictator game* there is no scope for such strategic considerations why a rational, self-interested individual would offer more as *proposer* in the *ultimatum game* relative to the transfer in *dictator game*. Considering whether *reflective* subjects are more likely to transfer DKK 0 in *dictator game*, have an acceptance threshold of DKK 0 or DKK 10 in the *ultimatum game*, exhibit strategic fairness as *proposer* in the *ultimatum game* and not opting for the *punishment* opportunity in the *third-party punishment task*, I find this to be the case. 20.9% of the *reflective* subjects complied with the above-mentioned as opposed to 6.3% of those not *reflective*. This difference is statistically significant ($p = 0.008$, $\chi^2 - test$). When controlling for gender, *reflective* subjects are predicted to be 12.2%-points more likely than *non-reflective* subjects to choose as described in these tasks. *Reflective* subjects are predicted to choose as described with a probability of 19.8% as opposed to a predicted probability of 7.6% for those *non-reflective* (see Table 1).

**Table 1.** Marginal effects from Logistic regressions.

| VARIABLES | (1) A | (2) B | (3) C | (4) D | (5) E | (6) F | (7) G |
|---|---|---|---|---|---|---|---|
| 1.Reflective | 0.098 | 0.126 *** | 0.114 | 0.040 | 0.201 *** | 0.108 *** | 0.122 ** |
| | (0.069) | (0.037) | (0.071) | (0.074) | (0.074) | (0.032) | (0.048) |
| 1.Male | 0.233 *** | 0.091 ** | 0.072 | 0.210 *** | 0.029 | 0.089 ** | 0.165 *** |
| | (0.062) | (0.042) | (0.065) | (0.065) | (0.065) | (0.038) | (0.048) |
| Observations | 235 | 235 | 235 | 235 | 235 | 235 | 235 |

Standard errors in parentheses. *** $p < 0.01$, ** $p < 0.05$, * $p < 0.1$. A: Dictator Game transfer = 0, B: Ultimatum Game offer = 0 or = 10, C: Ultimatum Game acceptance threshold = 0 or = 10, D: Strategic fairness; Ultimatum Game offer greater than Dictator Game transfer, E: Punishment sacrifice = 0, F: Compliance with A; B; C; E, G: Compliance with A; C; D; E.

## 5. Discussion

In line with several other studies, this study found more rational, self-interested behavior among more *reflective* individuals and more prosocial behavior among *intuitive* individuals. Further, this study found the more prosocial behavior among *intuitive* individuals to carry over to the *third-party punishment task*, where these individuals were found more likely to sanction a selfish act. A contribution of the present study was that subjects were to complete multiple social dilemma task, which allows to investigate the consistency across choices. In this aspect, *reflective* individuals were found more likely to act rationally in accordance with their self-interest across all four decisions.

*Intuitive* subjects give more in the *dictator game*, which is consistent with the findings of [7]. Transferring a positive amount to the *receiver* in the *dictator game* could be interpreted as altruistic preferences [2]. However, the findings of more rational and self-interested behavior by *reflective* subjects should be interpreted carefully, as gender seems to be the significant factor that drives differences in behavior in the *dictator game*. This is consistent with the findings of women giving more in a meta study on the experiments testing for gender differences [18].

In the *ultimatum game*, *reflective* subjects offered less than those not *reflective*. The decision of the *proposer* can be explained either by a "taste for fairness" or a "fear of rejection" (or a combination of these motives) [22]. Including the *dictator game* allows the inference with which motive matters for which group. However, the results indicate that both groups seem to act on a "fear of rejection". These findings contradict the findings of difference in transfer/offer being driven mostly by *reflective* individuals [10]. Even though "strategic fairness" appears to exist among both groups, the offers of the *intuitive* individuals are larger than those of the *reflective*. Thus, *intuitive* individuals appear to expect their offers in the *ultimatum game* to more likely be rejected. This is consistent with the consensus effect [23]. *Intuitive* individuals require a larger amount to accept an offer themselves.

*Reflective* subjects are more likely to accept offers in the *ultimatum game*, which confirms the findings of [8,9]. In those studies, the "strategy method" was not applied to the recipient's decision. Thus, *reflective* individuals exhibit a greater willingness to accept an unfair *ultimatum game* offer even when they are not directly faced with and possibly offended by the offer. Whether or not the strategic version of the *ultimatum game* induces lower acceptance thresholds is to some degree addressed in [24]. In this study, besides from playing the extensive form of the game, the subjects were required to state the minimum offer she would be willing accept. They found a significant negative correlation between the acceptance threshold and proposed offer which can be interpreted in light of *reflective* behavior. These individuals understand the bargaining position of the game as well as the risk of being rejected. Considering "negative reciprocity" as the motive for rejecting unfair offers in the *ultimatum game*, *reflective* individuals are more capable of overcoming their *intuitive* desire to *punish* the selfish act by the *proposer*. The willingness to accept an unfair offer is related to the ability to reflect further upon the decision and realize that accepting the offer is the better option.

*Intuitive* subjects are more likely to engage in *third-party punishment* and *reflective* subjects appear again more likely to act rational and self-interested. Thus, *intuitive* individuals are interpreted to be more likely to act reciprocally.

## 6. Limitations

Some factors related to the experimental design may have influenced how subjects behaved.

As the link to the survey were distributed at lectures encouraging students to participate, it is unknown when, where and possibly with whom the subjects completed the survey. Hence, there is concerns regarding their anonymity. Considering the relatively high share of correct answers in the CRT, one could expect subjects to have communicated with each other or have accessed the internet to look up the correct answer. Further, the chances of receiving payment for completing the CRT depended on the number of correct answers, which might have further incentivized subjects to look up the correct answer - at least incentivized them to think more carefully about the question, which was unintended. These limitations question whether the categorization of subjects is reliable. A reasonable explanation for the relatively high share of correct answers on the CRT in this study is the test's correlation with math abilities [5]. The vast majority of subjects were students of Economics, Political Science or Psychology. Especially students of Economics are expected to be relatively more capable of math. The survey questions did not elicit from which education the subjects were enrolled.

Only seven of the 295 subjects who completed the study received payment, providing only weak incentives. However, the observations here fit rather well the observations from other studies with stronger economic incentives. In a meta study, the average transfer was found to be 28% of the endowment [18], which is not far from 30.4% observed in this study. In a meta study on the ultimatum game, subjects were found to offer 40% of the endowment on average [17], which is comparable to the 43.5% observed here.

Further, around 20% of the subjects who started completing the survey opted out before the final question. Not being able to control the condition under which the survey was completed increases the probability of subjects sabotaging the experiment by choosing randomly or not reading through the

instructions thoroughly. However, including or excluding the "outliers" of the present study did not change results.

## 7. Concluding Remarks

*Reflective* individuals are more likely to act rational and self-interested in social dilemma tasks and *intuitive* individuals are more likely to bring their internalized cooperative and fair behavior to the lab. Acknowledging that individuals differ in their cognitive reflection ability entails greater prediction and description of decision making. *Intuitive* individuals are more likely to act as a strong reciprocator and do not tolerate selfish deviations for material incentives. Explaining the *intuitive* decision in the lab by the Social Heuristic Hypothesis insights are gained regarding how society maintains the cooperative and fair behavior and could shed light on cultural differences. A topic for future research is to investigate whether the *intuitive* behavior is prosocial across cultures. Future research could differentiate the perspectives further to predict decision making with greater precision and understand the behavioral differences in more detail.

**Funding:** This research received no external funding.

**Acknowledgments:** I want to thank Alexander Koch for helpful comments and feedback during the process of designing the experiments as well as writing the article.

**Conflicts of Interest:** The author declares no conflict of interest.

## Appendix A. Survey Instructions

Q1: I would really appreciate your help in collecting data for my bachelor thesis. Completing this survey will only take a few minutes and you will have the chance to earn up to DKK 400 by answering seven survey questions. I will randomly draw 7 participants, who will get paid according to their choices. This will be explained in the survey. My name is Markus Seier and I am studying Economics. Your participation is voluntary. I will analyze the data in anonymous format. The email address that you can provide at the end of the survey will only be used to contact you in case you are among the participants drawn to receive a payment. Payments will be made by mobile pay. I will delete the email address as soon as payments are completed.

Q2: First, you complete four tasks regarding "division of money". Your decisions in these tasks determine your earnings if you are randomly drawn to be paid for answering this survey. If you are drawn to be paid for a particular question, you are paid according to your choices and the choices of the other participants with whom you are randomly matched. You can be drawn to be paid for multiple questions. After completing the four above-mentioned tasks, you proceed to the second part of this survey with three short questions. Lastly, you are to indicate your gender, at which faculty you study and provide your email address if you want to have a chance of getting paid up to DKK 400. Please continue to the next page where you are to complete four different tasks regarding division of money.

Q3: You are matched with another participant of this survey. You are given DKK 100 and must decide on how much to offer the other participant. You act as the "proposer". You earn DKK 100 subtracted what you have offered and the other participant earns what you have offered him/her. How much do you give to the other participant? Remember that you and the other participant will actually be paid according to your decisions if the computer draws your names.

- DKK 0 (That is: You get DKK 100. The other gets DKK 0.)
- DKK 10 (That is: You get DKK 90. The other gets DKK 10.)
- DKK 20 (That is: You get DKK 80. The other gets DKK 20.)
- DKK 30 (That is: You get DKK 70. The other gets DKK 30.)
- DKK 40 (That is: You get DKK 60. The other gets DKK 40.)
- DKK 50 (That is: You get DKK 50. The other gets DKK 50.)
- DKK 60 (That is: You get DKK 40. The other gets DKK 60.)

- DKK 70 (That is: You get DKK 30. The other gets DKK 70.)
- DKK 80 (That is: You get DKK 20. The other gets DKK 80.)
- DKK 90 (That is: You get DKK 10. The other gets DKK 90.)
- DKK 100 (That is: You get DKK 0. The other gets DKK 100.)

Q4: You are matched with another participant of this survey. You are given DKK 100 and must decide on how much to offer the other participant. If the other participant accepts your offer, you earn DKK 100 subtracted what you have offered and the other participant earns what you have offered him/her. If the other participant rejects your offer, you both earn DKK 0. How much do your offer the other participant? Remember that you and the other participant will actually be paid according to your decisions if the computer draws your names.

- DKK 0 (That is: You get DKK 100, The other gets DKK 0 if the offer is accepted. Otherwise, you both get DKK 0.)
- DKK 10 (That is: You get DKK 90, The other gets DKK 10 if the offer is accepted. Otherwise, you both get DKK 0.)
- DKK 20 (That is: You get DKK 80, The other gets DKK 20 if the offer is accepted. Otherwise, you both get DKK 0.)
- DKK 30 (That is: You get DKK 70, The other gets DKK 30 if the offer is accepted. Otherwise, you both get DKK 0.)
- DKK 40 (That is: You get DKK 60, The other gets DKK 40 if the offer is accepted. Otherwise, you both get DKK 0.)
- DKK 50 (That is: You get DKK 50, The other gets DKK 50 if the offer is accepted. Otherwise, you both get DKK 0.)
- DKK 60 (That is: You get DKK 40, The other gets DKK 60 if the offer is accepted. Otherwise, you both get DKK 0.)
- DKK 70 (That is: You get DKK 30, The other gets DKK 70 if the offer is accepted. Otherwise, you both get DKK 0.)
- DKK 80 (That is: You get DKK 10, The other gets DKK 80 if the offer is accepted. Otherwise, you both get DKK 0.)
- DKK 90 (That is: You get DKK 10, The other gets DKK 90 if the offer is accepted. Otherwise, you both get DKK 0.)
- DKK 100 (That is: You get DKK 0, The other gets DKK 100 if the offer is accepted. Otherwise, you both get DKK 0.)

Q5: You must now decide whether to accept or reject an offer from another participant. The other participant is given DKK 100 and must decide on how much to offer you. If you accept, you earn what the other participant offered you and the other participant earns DKK 100 subtracted what he/she offered you. If you reject, you both earn DKK 0. What is the minimum offer, you are willing to accept? Remember that you and the other participant will actually be paid according to your decisions if the computer draws your names.

- DKK 0
- DKK 10
- DKK 20
- DKK 30
- DKK 40
- DKK 50
- DKK 60
- DKK 70
- DKK 80

- DKK 90
- DKK 100

Q6: I will randomly draw a pair of participants, from the first question, were the participant endowed with DKK 100 (the "proposer") chose to give DKK 0 and keep the DKK 100 for him/herself. You are given DKK 50 and can reduce the earnings of the proposer who chose to keep the DKK 100 for him/herself. You can reduce the earnings of the proposer by DKK 5 by giving up DKK 1 of your own earnings. That is, if you give up DKK X of your own earnings, you reduce the earnings of the proposer by DKK 5*X. How much of your own earnings are you willing to give up to reduce the earnings of the proposer? Remember that you and the other participant will actually be paid according to your decisions if the computer draws your names.

- DKK 0 (Reduce the earnings of the proposer by DKK 0)
- DKK 1 (Reduce the earnings of the proposer by DKK 5)
- DKK 2 (Reduce the earnings of the proposer by DKK 10)
- DKK 3 (Reduce the earnings of the proposer by DKK 15)
- DKK 4 (Reduce the earnings of the proposer by DKK 20)
- DKK 5 (Reduce the earnings of the proposer by DKK 25)
- DKK 6 (Reduce the earnings of the proposer by DKK 30)
- DKK 7 (Reduce the earnings of the proposer by DKK 35)
- DKK 8 (Reduce the earnings of the proposer by DKK 40)
- DKK 9 (Reduce the earnings of the proposer by DKK 45)
- DKK 10 (Reduce the earnings of the proposer by DKK 50)
- DKK 11 (Reduce the earnings of the proposer by DKK 55)
- DKK 12 (Reduce the earnings of the proposer by DKK 60)
- DKK 13 (Reduce the earnings of the proposer by DKK 65)
- DKK 14 (Reduce the earnings of the proposer by DKK 70)
- DKK 15 (Reduce the earnings of the proposer by DKK 75)
- DKK 16 (Reduce the earnings of the proposer by DKK 80)
- DKK 17 (Reduce the earnings of the proposer by DKK 85)
- DKK 18 (Reduce the earnings of the proposer by DKK 90)
- DKK 19 (Reduce the earnings of the proposer by DKK 95)
- DKK 20 (Reduce the earnings of the proposer by DKK 100)

You have now completed the first part of the survey. The next part consists of three questions, where you are to write your answer in the box below the question. Your chances of getting paid for this part depend on how many questions you answer correctly. For each correct answer, one lottery ticket with your name will be added to the pool from which the computer will draw one participant, who will be paid DKK 100.

Q7: A bat and a ball cost $1.10. The bat costs $1.00 more than the ball. How much does the ball cost? (Write your answer in cents) Remember, a correct answer increases your chances of getting paid DKK 100.

Q8: If it takes 5 machines 5 minutes to make 5 widgets, how long would it take 100 machines to make 100 widgets? (Write your answer in minutes) Remember, a correct answer increases your chances of getting paid DKK 100.

Q9: In a lake, there is a patch of lily pads. Every day, the patch doubles in size. If it takes 48 days for the patch to cover the entire lake, how long would it take for the patch to cover half of the lake? (Write your answer in days) Remember, a correct answer increases your chances of getting paid DKK 100.

Q10: Please indicate your gender.

- Male
- Female

    Q11: At which faculty do you study?

- Arts
- Health
- Science & Technology
- BSS

    Q12: Please write your email-address (studynumber@post.au.dk) The email address is to pay a participant who is drawn to receive his/her earnings in the survey. You are not required to provide your email address, but you cannot get paid if you do not.

    Thank you for participating. You will be notified by email if you are drawn to be paid.

**Appendix B. Cognitive Reflection Test**

1.  A bat and a ball cost $1.10. The bat costs $1.00 more than the ball. How much does the ball cost? ___ cents. Intuitive Answer: 10 / Correct Answer: 5.
2.  If it takes 5 machines 5 minutes to make 5 widgets, how long would it take 100 machines to make 100 widgets? _____ minutes. Intuitive Answer: 100 / Correct Answer: 5.
3.  In a lake, there is a patch of lily pads. Every day, the patch doubles in size. If it takes 48 days for the patch to cover the entire lake, how long would it take for the patch to cover half of the lake?_____ days. Intuitive Answer: 24 / Correct Answer: 47.

**Appendix C. Cognitive Reflection Test Categorization**

$$Intuitive \begin{cases} = 1 & if\ Q1 = 10\ \&\ Q2 = 100\ or\ Q1 = 10\ \&\ Q3 = 24\ or\ Q2 = 100\ \&\ Q3 = 24 \\ = 0 & Otherwise \end{cases}$$

$$Reflective \begin{cases} = 1 & if\ Q1 = 5\ \&\ Q2 = 5\ or\ Q1 = 5\ \&\ Q3 = 47\ or\ Q2 = 5\ \&\ Q3 = 47 \\ = 0 & Otherwise \end{cases}$$

$$Residual \begin{cases} = 1 & if\ Intuitive = 0\ \&\ Reflective = 0 \\ = 0 & Otherwise \end{cases}.$$

**Appendix D. Results Excluding the Residual Group**

*Appendix D.1. Rational and Self-Interested Behavior*

    A graphical representation of decisions consistent with rational, self-interested behavior by *intuitive (reflective)* individuals can be found in Figure A1.

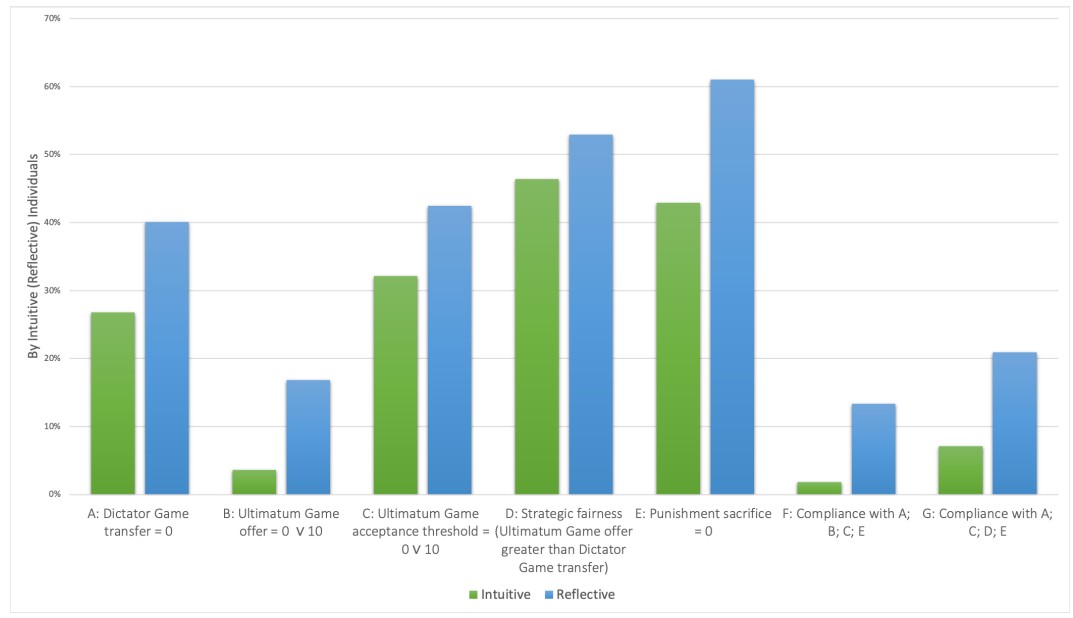

**Figure A1.** Frequency of Decision by Intuitive (Reflective) Individuals.

*Appendix D.2. Transfer in the Dictator Game*

The distribution of the *dictator game* transfer can be found in Table A1 and is illustrated in Figure A2.

**Table A1.** Frequency of Dictator Game Transfer by Intuitive (Reflective) Individuals.

| Dictator Game Transfer | Intuitive | Reflective | Total |
|---|---|---|---|
| Transfer = 0 | 26.6% | 40.12% | 36.84% |
| Transfer = 10 | 1.8% | 1.16% | 1.32% |
| Transfer = 20 | 3.57% | 4.65% | 4.39% |
| Transfer = 30 | 3.57% | 2.91% | 3.07% |
| Transfer = 40 | 8.93% | 5.81% | 6.58% |
| Transfer = 50 | 48.21% | 41.86% | 43.42% |
| Transfer = 60 | 1.79% | 1.16% | 1.32% |
| Transfer = 70 | 0% | 0% | 0% |
| Transfer = 80 | 1.79% | 0% | 0.44% |
| Transfer = 90 | 0% | 0% | 0% |
| Transfer = 100 | 3.57% | 2.33% | 2.63% |

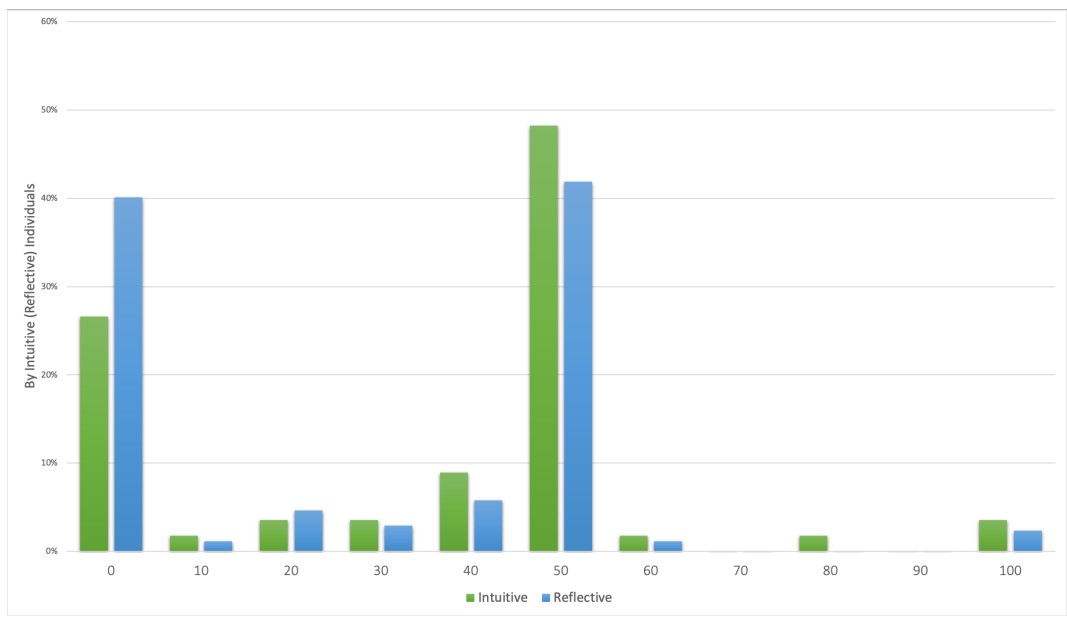

**Figure A2.** Frequency of Transfer by Intuitive (Reflective) Individuals.

*Appendix D.3. Proposer Behavior in the Ultimatum Game*

The distribution of the proposer decision in the *ultimatum game* can be found in Table A2 and is illustrated in Figure A3.

**Table A2.** Frequency of Ultimatum Game Offer by Intuitive (Reflective) Individuals.

| Ultimatum Game Offer | Intuitive | Reflective | Total |
|:---:|:---:|:---:|:---:|
| Offer = 0 | 1.79% | 1.16% | 1.32% |
| Offer = 10 | 1.79% | 15.70% | 12.28% |
| Offer = 20 | 0% | 4.07% | 3.07% |
| Offer = 30 | 1.79% | 4.65% | 3.95% |
| Offer = 40 | 3.57% | 7.56% | 6.58% |
| Offer = 50 | 82.14% | 63.37% | 67.98% |
| Offer = 60 | 3.57% | 2.33% | 2.63% |
| Offer = 70 | 0% | 0% | 0% |
| Offer = 80 | 0% | 0.58% | 0.44% |
| Offer = 90 | 0% | 0% | 0% |
| Offer = 100 | 5.36% | 0.58% | 1.75% |

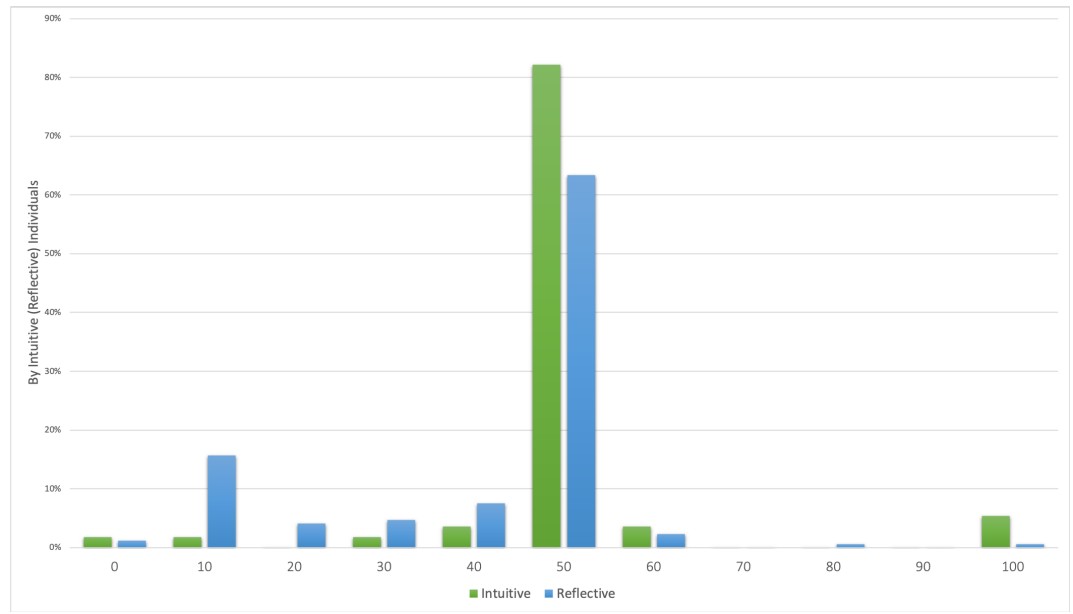

**Figure A3.** Frequency of Ultimatum Game Offer by Intuitive (Reflective) Individuals.

*Appendix D.4. Recipient Behavior in the Ultimatum Game*

The distribution of the recipient decision in the *ultimatum game* can be found in Table A3 and is illustrated in Figure A4.

**Table A3.** Frequency of Ultimatum Game Acceptance Threshold by Intuitive (Reflective) Individuals.

| Ultimatum Game Acceptance Threshold | Intuitive | Reflective | Total |
|:---:|:---:|:---:|:---:|
| Threshold = 0 | 10.71% | 5.81% | 7.02% |
| Threshold = 10 | 21.42% | 36.63% | 32.89% |
| Threshold = 20 | 1.79% | 5.23% | 4.39% |
| Threshold = 30 | 10.71% | 9.30% | 9.65% |
| Threshold = 40 | 16.07% | 16.86% | 16.67% |
| Threshold = 50 | 37.50% | 25.00% | 28.07% |
| Threshold = 60 | 0% | 0% | 0% |
| Threshold = 70 | 0% | 0% | 0% |
| Threshold = 80 | 0% | 0% | 0% |
| Threshold = 90 | 0% | 1.16% | 0.88% |
| Threshold = 100 | 1.79% | 0% | 0.44% |

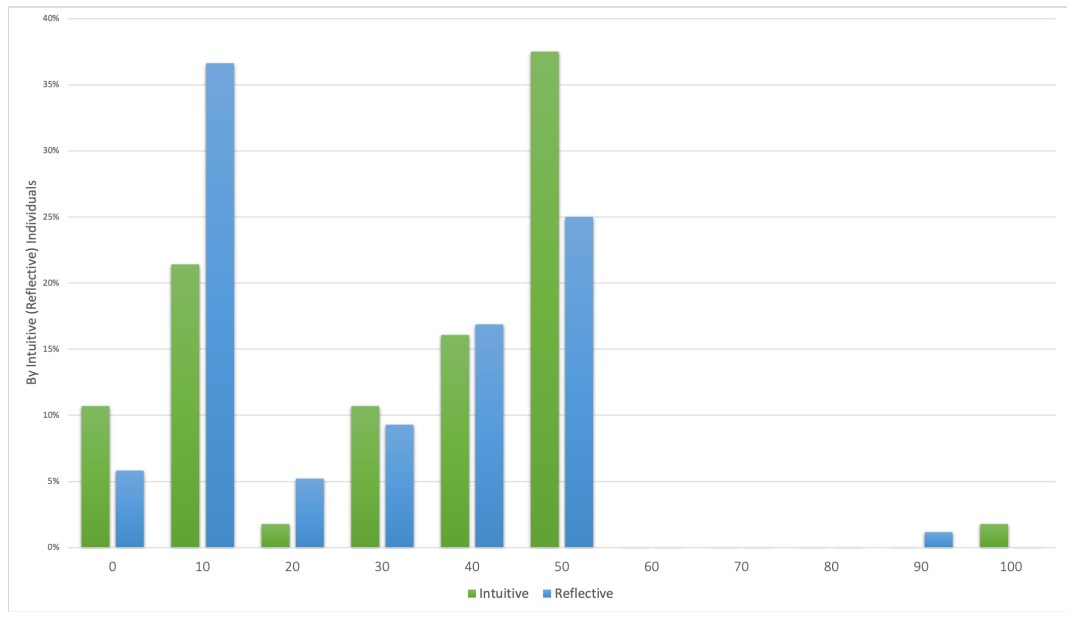

**Figure A4.** Frequency of Ultimatum Game Acceptance Threshold by Intuitive (Reflective) Individuals.

*Appendix D.5. Third-Party Punishment Behavior*

The distribution of the *third-party punishment* decision can be found in Table A4 and is illustrated in Figure A5.

**Table A4.** Frequency of Punishment Sacrifice by Intuitive (Reflective) Individuals.

| Punishment Sacrifice | Intuitive | Reflective | Total |
|:---:|:---:|:---:|:---:|
| Sacrifice = 0 | 42.86% | 61.05% | 56.58% |
| Sacrifice = 1 | 0% | 1.74% | 1.32% |
| Sacrifice = 2 | 1.79% | 0.58% | 0.88% |
| Sacrifice = 3 | 1.79% | 1.16% | 1.32% |
| Sacrifice = 4 | 3.57% | 1.16% | 1.75% |
| Sacrifice = 5 | 8.93% | 6.40% | 7.02% |
| Sacrifice = 6 | 0% | 1.16% | 0.88% |
| Sacrifice = 7 | 0% | 0.58% | 0.44% |
| Sacrifice = 8 | 0% | 1.16% | 0.88% |
| Sacrifice = 9 | 0% | 0% | 0% |
| Sacrifice = 10 | 19.64% | 13.95% | 15.35% |
| Sacrifice = 11 | 0% | 0.58% | 0.44% |
| Sacrifice = 12 | 1.79% | 1.16% | 1.32% |
| Sacrifice = 13 | 0% | 1.16% | 0.88% |
| Sacrifice = 14 | 0% | 0% | 0% |
| Sacrifice = 15 | 1.79% | 0.58% | 0.88% |
| Sacrifice = 16 | 0% | 0% | 0% |
| Sacrifice = 17 | 0% | 0.58% | 0.44% |
| Sacrifice = 18 | 0% | 0% | 0% |
| Sacrifice = 19 | 0% | 0% | 0% |
| Sacrifice = 20 | 17.86% | 6.98% | 9.65% |

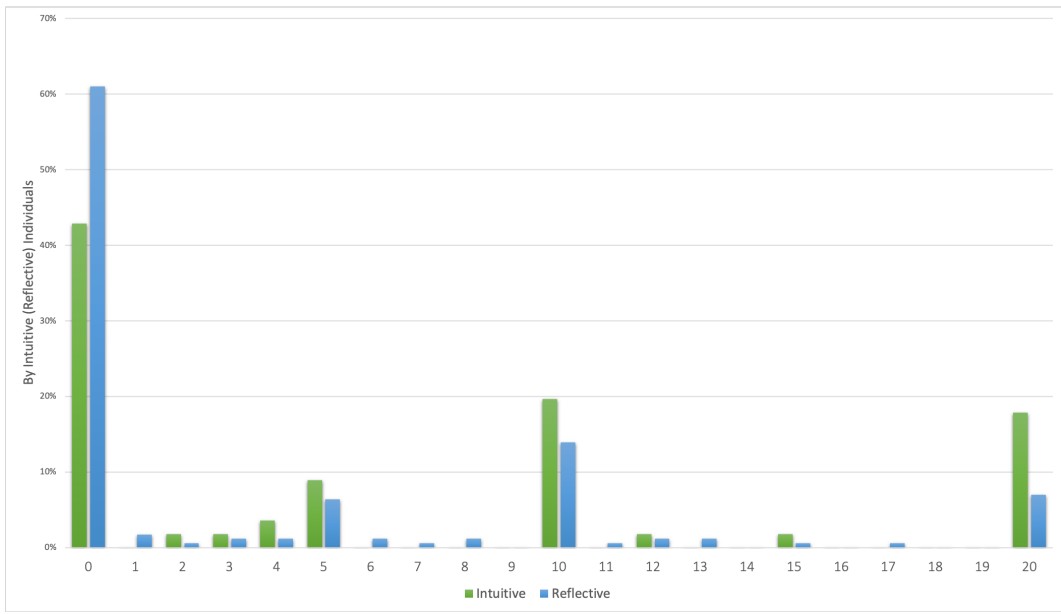

**Figure A5.** Frequency of Punishments Sacrifice by Intuitive (Reflective) Individuals.

*Appendix D.6. Behavioral Differences between Intuitive and Reflective Individuals (Excluding Residual Group)*

In Table A5 an overview of the results when excluding the residual group can be found. This include means of the different tasks as well as *p*-values from the statistical tests. A table with the marginal effects from logistic regressions can be found in Table A6. Excluding the residual group from the analyses and comparing those categorized as *reflective* only with those categorized as *intuitive* does not change much in the conclusions. Most notable differences are in terms of statistical significant in the MWU distribution tests and the contingency-table $\chi^2$ tests where the *p*-values are greater for almost all of the tasks. The logistic regressions excluding the residual group reveal a very similar pattern in terms of statistical significant and interpretation of marginal effects.

**Table A5.** Results Excluding the Residual Group by Intuitive (Reflective) Individuals.

|  | Intuitive | Reflective | Combined | MWU or $\chi^2$ (*p*-Value) |
|---|---|---|---|---|
| Dictator Game Transfer (mean) | 35.7 | 28.2 | 30 | 0.074 |
| A: Dictator Game Transfer = 0 (freq.) | 26.8% | 40.1% | 36.8% | 0.072 |
| Ultimatum Game Offer (mean) | 50.78 | 40.9 | 43.3 | 0.000 |
| B: Ultimatum Game Offer = 0 ∨ 10 (freq.) | 3.57% | 16.9% | 13.7% | 0.012 |
| Ultimatum Game Acceptance Threshold (mean) | 32.7 | 27.8 | 29 | 0.127 |
| C: Ultimatum Game Acceptance Threshold = 0 ∨ 10 (freq.) | 32.1% | 42.4% | 39.9% | 0.172 |
| D: Strategic Fairness (Dictator Game Transfer > Ultimatum Game Offer) (freq.) | 46.4% | 52.9% | 51.3% | 0.400 |
| Punishment Sacrifice (mean) | 6.7 | 4 | 4.6 | 0.010 |
| E: Punishment Sacrifice = 0 (freq.) | 42.9% | 61.1% | 56.6% | 0.017 |
| Compliance with A, B, C & E | 1.8% | 13.4% | 10.5% | 0.014 |
| Compliance with A, C, D & E | 7.14% | 20.9% | 17.5% | 0.018 |

**Table A6.** Marginal effects from Logistic regressions.

| VARIABLES | (1)<br>A | (2)<br>B | (3)<br>C | (4)<br>D | (5)<br>E | (6)<br>F | (7)<br>G |
|---|---|---|---|---|---|---|---|
| 1.Reflective | 0.081 | 0.123 *** | 0.090 | 0.014 | 0.177 ** | 0.107 *** | 0.114 ** |
| | (0.072) | (0.040) | (0.075) | (0.076) | (0.078) | (0.033) | (0.051) |
| 1.Male | 0.242 *** | 0.094 ** | 0.076 | 0.222 *** | 0.036 | 0.092 ** | 0.169 *** |
| | (0.063) | (0.043) | (0.066) | (0.066) | (0.067) | (0.039) | (0.049) |
| Observations | 227 | 227 | 227 | 227 | 227 | 227 | 227 |

Standard errors in parentheses. *** $p < 0.01$, ** $p < 0.05$, * $p < 0.1$. A: Dictator Game transfer = 0, B: Ultimatum Game offer = 0 or = 10, C: Ultimatum Game acceptance threshold = 0 or = 10, D: Strategic fairness; Ultimatum Game offer greater than Dictator Game transfer, E: Punishment sacrifice = 0, F: Compliance with A; B; C; E, G: Compliance with A; C; D; E.

## Appendix E. Cognitive Reflection Test Results

The distribution of answers on the CRT for both men and women can be found in Table A7, for men alone in Table A8 and for women in Table A9.

**Table A7.** Distribution of Answers on the CRT for Both Men and Women.

| Question/Answer | Correct | Intuitive | Other |
|---|---|---|---|
| 1: Bat and Ball | 58% | 39% | 3% |
| 2: Widget | 69% | 25% | 6% |
| 3: Lily Pads | 82% | 15% | 3% |

**Table A8.** Distribution of Answers on the CRT for Men Alone.

| Question/Answer | Correct | Intuitive | Other |
|---|---|---|---|
| 1: Bat and Ball | 65% | 31% | 3% |
| 2: Widget | 77% | 20% | 3% |
| 3: Lily Pads | 89% | 10% | 1% |

**Table A9.** Distribution of Answers on the CRT for Women Alone.

| Question/Answer | Correct | Intuitive | Other |
|---|---|---|---|
| 1: Bat and Ball | 50% | 48% | 2% |
| 2: Widget | 60% | 30% | 10% |
| 3: Lily Pads | 74% | 21% | 5% |

The distribution of the number of correct answers on the CRT by gender and in total can be found in Table A10 and the distribution of the number of intuitive, wrong answers can be found in Table A11.

**Table A10.** Distribution of Number of Correct Answers on the CRT by Gender.

| Gender/Number of<br>Correct Answers | 0 Correct<br>Answers | 1 Correct<br>Answer | 2 Correct<br>Answers | 3 Correct<br>Answers |
|---|---|---|---|---|
| Men | 5% | 13% | 27% | 55% |
| Women | 21% | 16% | 22% | 41% |
| Men & Women | 13% | 14% | 24% | 49% |

**Table A11.** Distribution of Number of Intuitive Answers on the CRT by Gender.

| Gender/Number of Intuitive Answers | 0 Intuitive Answers | 1 Intuitive Answer | 2 Intuitive Answers | 3 Intuitive Answers |
|---|---|---|---|---|
| Men | 60% | 23% | 12% | 5% |
| Women | 47% | 22% | 18% | 13% |
| Men & Women | 54% | 22% | 15% | 9% |

## Appendix F. Additional Tables and Histograms of Choices by Non-Reflective (Reflective) Individuals

*Appendix F.1. Transfer in the Dictator Game*

The distribution of the *dictator game* transfer can be found in Table A12 and is illustrated in Figure A6.

**Table A12.** Frequency of Dictator Game Transfer by Non-Reflective (Reflective) Individuals.

| Dictator Game Transfer | Non-Reflective | Reflective | Total |
|---|---|---|---|
| Transfer = 0 | 25% | 40.12% | 36.02% |
| Transfer = 10 | 1.56% | 1.16% | 1.27% |
| Transfer = 20 | 3.12% | 4.65% | 4.24% |
| Transfer = 30 | 4.69% | 2.91% | 3.39% |
| Transfer = 40 | 7.81% | 5.81% | 6.36% |
| Transfer = 50 | 51.56% | 41.86% | 44.49% |
| Transfer = 60 | 1.56% | 1.16% | 1.27% |
| Transfer = 70 | 0% | 0% | 0% |
| Transfer = 80 | 1.56% | 0% | 0.42% |
| Transfer = 90 | 0% | 0% | 0% |
| Transfer = 100 | 3.12% | 2.33% | 2.54% |

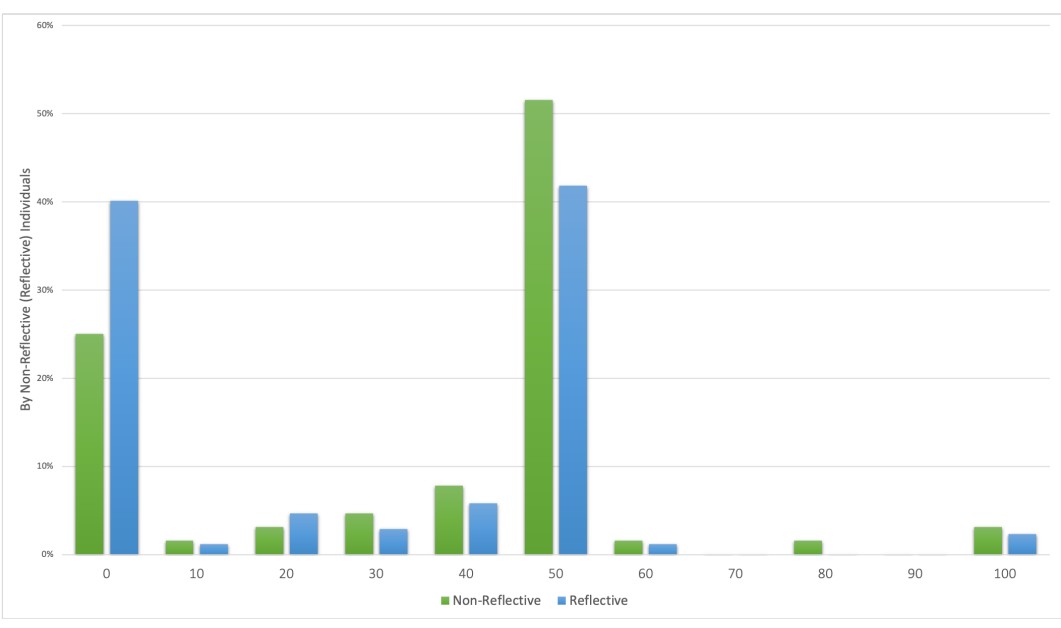

**Figure A6.** Frequency of Transfer by Non-Reflective (Reflective) Individuals.

*Appendix F.2. Proposer Behavior in the Ultimatum Game*

The distribution of the proposer decision in the *ultimatum game* can be found in Table A13 and is illustrated in Figure A7.

**Table A13.** Frequency of Ultimatum Game Offer by Non-Reflective (Reflective) Individuals.

| Ultimatum Game Offer | Non-Reflective | Reflective | Total |
|:---:|:---:|:---:|:---:|
| Offer = 0 | 1.56% | 1.16% | 1.27% |
| Offer = 10 | 1.56% | 15.70% | 11.86% |
| Offer = 20 | 0% | 4.07% | 2.97% |
| Offer = 30 | 1.56% | 4.65% | 3.81% |
| Offer = 40 | 4.69% | 7.56% | 6.78% |
| Offer = 50 | 82.81% | 63.37% | 68.64% |
| Offer = 60 | 3.12% | 2.33% | 2.54% |
| Offer = 70 | 0% | 0% | 0% |
| Offer = 80 | 0% | 0.58% | 0.42% |
| Offer = 90 | 0% | 0% | 0% |
| Offer = 100 | 4.69% | 0.58% | 1.69% |

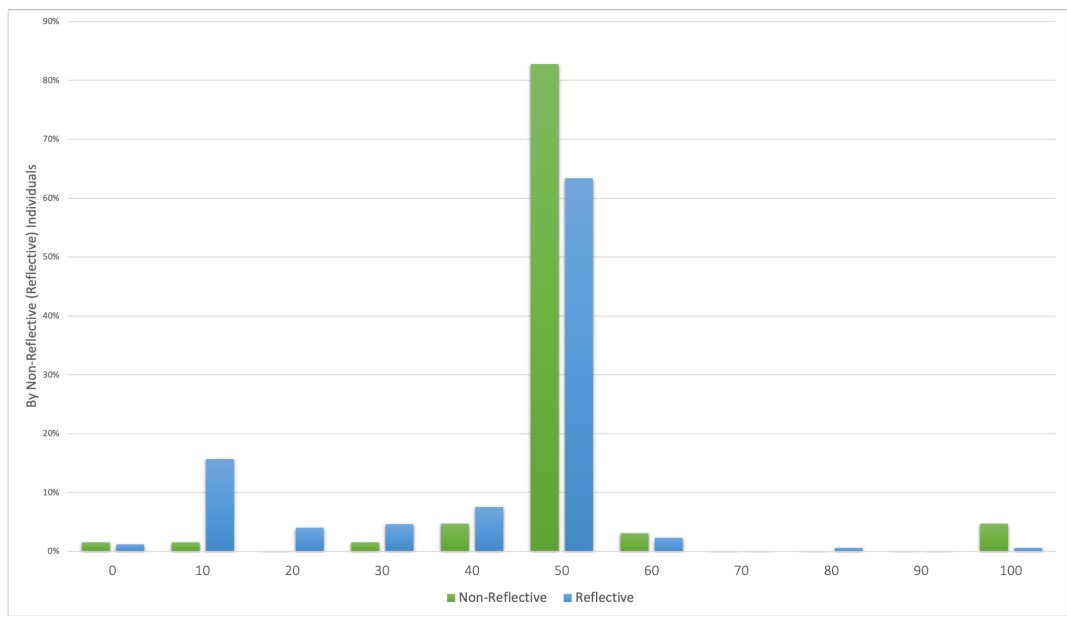

**Figure A7.** Frequency of Ultimatum Game Offer by Non-Reflective (Reflective) Individuals.

*Appendix F.3. Recipient Behavior in the Ultimatum Game*

The distribution of the recipient decision in the *ultimatum game* can be found in Table A14 and is illustrated in Figure A8.

**Table A14.** Frequency of Ultimatum Game Acceptance Threshold by Non-Reflective (Reflective) Individuals.

| Ultimatum Game Acceptance Threshold | Non-Reflective | Reflective | Total |
|---|---|---|---|
| Threshold = 0 | 9.38% | 5.81% | 6.78% |
| Threshold= 10 | 20.31% | 36.63% | 32.20% |
| Threshold= 20 | 1.56% | 5.23% | 4.24% |
| Threshold= 30 | 10.94% | 9.30% | 9.75% |
| Threshold= 40 | 14.06% | 16.86% | 16.10% |
| Threshold= 50 | 42.19% | 25.00% | 29.66% |
| Threshold= 60 | 0% | 0% | 0% |
| Threshold= 70 | 0% | 0% | 0% |
| Threshold= 80 | 0% | 0% | 0% |
| Threshold= 90 | 0% | 1.16% | 0.85% |
| Threshold= 100 | 1.56% | 0% | 0.42% |

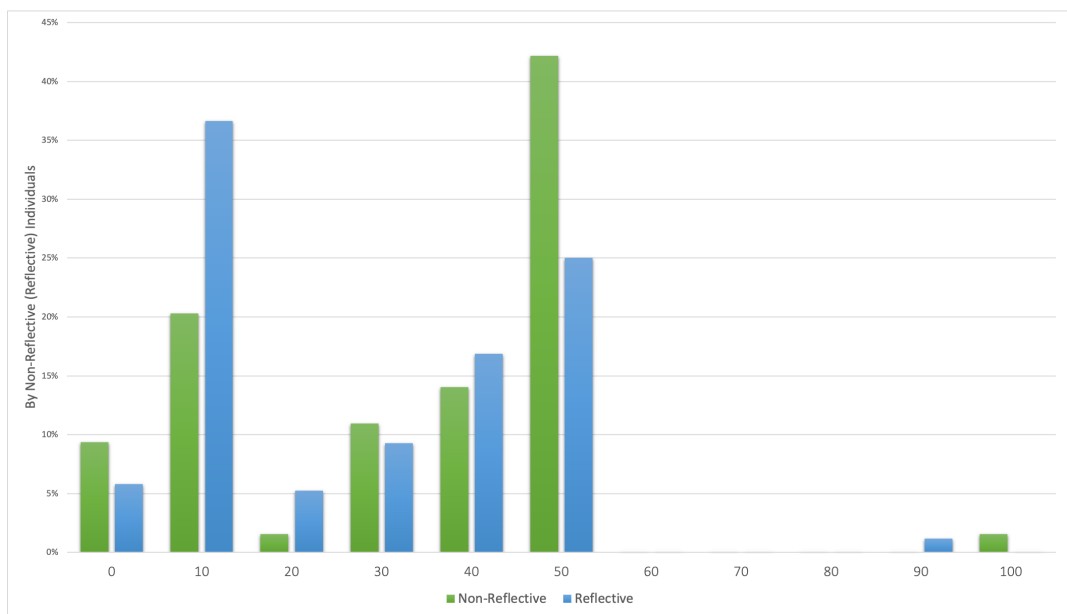

**Figure A8.** Frequency of Ultimatum Game Acceptance Threshold by Non-Reflective (Reflective) Individuals.

*Appendix F.4. Third-Party Punishment Behavior*

The distribution of the *third-party punishment* decision can be found in Table A15 and is illustrated in Figure A9.

**Table A15.** Frequency of Punishment Sacrifice by Non-Reflective (Reflective) Individuals.

| Punishment Sacrifice | Non-Reflective | Reflective | Total |
|---|---|---|---|
| Sacrifice = 0 | 40.62% | 61.05% | 55.51% |
| Sacrifice = 1 | 0% | 1.74% | 1.27% |
| Sacrifice = 2 | 1.56% | 0.58% | 0.85% |
| Sacrifice = 3 | 1.56% | 1.16% | 1.27% |
| Sacrifice = 4 | 4.69% | 1.16% | 2.12% |
| Sacrifice = 5 | 7.81% | 6.40% | 6.78% |
| Sacrifice = 6 | 0% | 1.16% | 0.85% |
| Sacrifice = 7 | 0% | 0.58% | 0.42% |
| Sacrifice = 8 | 0% | 1.16% | 0.85% |
| Sacrifice = 9 | 0% | 0% | 0% |
| Sacrifice = 10 | 18.75% | 13.95% | 15.25% |
| Sacrifice = 11 | 1.56% | 0.58% | 0.85% |
| Sacrifice = 12 | 3.12% | 1.16% | 1.69% |
| Sacrifice = 13 | 0% | 1.16% | 0.85% |
| Sacrifice = 14 | 0% | 0% | 0% |
| Sacrifice = 15 | 1.56% | 0.58% | 0.85% |
| Sacrifice = 16 | 0% | 0% | 0% |
| Sacrifice = 17 | 0% | 0.58% | 0.42% |
| Sacrifice = 18 | 0% | 0% | 0% |
| Sacrifice = 19 | 0% | 0% | 0% |
| Sacrifice = 20 | 18.75% | 6.98% | 10.17% |

**Figure A9.** Frequency of Punishment Sacrifice by Non-Reflective (Reflective) Individuals

*Appendix F.5. Behavioral Differences between Non-Reflective and Reflective Individuals*

In Table A16 an overview of the results comparing *non-reflective* and *reflective* individuals can be found.

**Table A16.** Results by Non-Reflective (Reflective) Individuals.

|  | Non-Reflective | Reflective | Combined | MWU or $\chi^2$ (*p*-Value) |
|---|---|---|---|---|
| Dictator Game Transfer (mean) | 36.4 | 28.2 | 30.4 | 0.033 |
| A: Dictator Game Transfer = 0 (freq.) | 25% | 40.1% | 36% | 0.032 |
| Ultimatum Game Offer (mean) | 50.5 | 40.9 | 43.5 | 0.000 |
| B: Ultimatum Game Offer = 0 ∨ 10 (freq.) | 3.1% | 16.9% | 13.1% | 0.005 |
| Ultimatum Game Acceptance Threshold (mean) | 33.9 | 27.8 | 29.4 | 0.032 |
| C: Ultimatum Game Acceptance Threshold = 0 ∨ 10 (freq.) | 29.7% | 42.4% | 39% | 0.074 |
| D: Strategic Fairness (Dictator Game Transfer > Ultimatum Game Offer) (freq.) | 43.8% | 52.9% | 50.4% | 0.211 |
| Punishment Sacrifice (mean) | 7.1 | 4 | 4.8 | 0.002 |
| E: Punishment Sacrifice = 0 (freq.) | 40.6% | 61.1% | 55.5% | 0.005 |
| Compliance with A, B, C & E | 1.6% | 13.4% | 10.2% | 0.008 |
| Compliance with A, C, D & E | 6.3% | 20.9% | 17% | 0.008 |

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
