# Peer review of "The Intuition of Punishment: A Study of Fairness Preferences and Cognitive Ability"

_games, doi:10.3390/g11020021_

Round 1
Reviewer 1 Report
Summary
The author conducts three well-known experimental paradigms on social preferences (Dictator game, ultimatum game, third party punishment game). Then, the author conducts a test that aims to measure if someone uses 'system 1' or 'system 2' thinking. The author correlates the findings of both measures, and finds that more 'rational' behavior in the experimental games correlates with system 2 thinking.
I like this study a lot. The experimental design is straightforward and I think it provides nice insights. Oftentimes in the literature response times are used to study to what degree subjects use intuitive or deliberate thinking. But, such an approach suffers from endogeneity issues (see Krajbich et al. 2015). As a small suggestion, I would encourage the author to very shortly discuss this work (two sentences or so). Other than that, I think this is a fine piece of work.
Literature
Krajbich, I., Bartling, B., Hare, T., & Fehr, E. (2015). Rethinking fast and slow based on a critique of reaction-time reverse inference. Nature communications, 6(1), 1-9.
Author Response
Please see the attachement.

Reviewer 2 Report
This article presents an analysis of the behavior of 236 students in the dictator and ultimatum games according whether a student is classified as being reflective or non-reflective. This classification was based on answers to a set of three questions where the "intuitive" answer was incorrect. A student was classified as being reflective when he/she answered at least two of the questions correctly.
On the basis of the results from these games, the author carries out a statistical analysis of the behaviour of the students using univariate tests and logistic regression. The author concludes that individuals classified as being reflective exhibit a higher level of self-interest in these games compared to those who are classified as not being reflective.
The games used are often used in study on behavioral economics. As such the subject of the study is of interest, although the study itself is not very innovative and the results are not particularly surprising or enlightening to those working in this area. However, the subject might well be of general interest to a wider range of readers interested in game theory.
I have a number of comments and suggestions to the author
- Comparisons between frequencies are often made in a confusing manner. For example, on l. 225-226, "Reflective subjects are estimated to be 12, 6% more likely than non-reflective subjects to offer DKK 0 or DKK 10 in the ultimatum game (see table 1)". This statement is ambiguous, since one might interpret this as a percentage change in the probability of such behavior (e.g. from 0.48 to 0.54 is a similar percentage change). The author should present these results in a less ambiguous manner, e.g. give estimates of the probability of the appropriate behavior in both groups.
-
2. L.187 onwards. The author first states that there are 9 individuals in the "residual" group, but in the next sentence states that there are 8 individuals in this group (based on the figures given, this second number seems to be correct). The author then states "As the residual group consists only of 8 subjects, these are included throughout the statistical analysis". Reading between the lines, it appears that these individuals are grouped together with the "intuitive" individuals to form a group of 64 non-reflective individuals. This should be stated explicitly.
- 3. L. 210-212 "However, a part of the difference can be contributed to gender: Males are found significantly more likely to transfer DKK 0 to the receiver in the dictator game (see table 1)". This statement needs to be substantiated, e.g. using multivariate logistic regression or analysis of a 3-d contingency table. Since "reflection" is clearly associated with both sex and behavior, it is possible that i) controlling for reflectiveness, behavior is independent of sex, ii) controlling for sex, behavior is independent of reflectiveness, or iii) behavior is associated with both explanatory variables (after controlling for the effect of the other).
- 4. L. 250-251 "Thus, both the reflective and intuitive subjects lower their transfer in the dictator game relative to their offer in the ultimatum game". Since in the experimental design the players choose their action in the dictator game before choosing their action as the proposer in the ultimatum game, it would be more natural to state "both reflective and intuitive subjects increase their transfer in the ultimatum game relative to their offer in the dictator game". The order in which the games are played in the study is natural. However, context (including the order in which games are presented) might play an important part in the behavior exhibited.
- 5. The author uses the "strategic version" of the ultimatum in which the respondent states the minimum offer that he/she would accept before actually receiving an offer. In the "extensive form version" of the ultimatum game, the respondent first receives an offer and then decides whether to accept it or not. Although in both versions of this game an economically rational respondent should accept any positive offer, the behavior exhibited in experimental games is significantly different. The probability of rejecting an offer is significantly higher under the "strategic version" than under the extensive form version. This difference very likely related to the level of reflection of the players. In the strategic form of the game, a player with reflective abilities might well, on the basis of analysing the game, express a willingness to accept a lower offer than 50% and demand a higher amount as a proposer. As a result one might expect that a reflective players would act similarly in both versions of the game. Less reflective players might require 50% in the strategic form of the game, but when faced with a concrete offer might reassess the situation and decide to accept a lower offer. This question is addressed (to some degree) in
- Markowska-Przybyła U. and Ramsey D. (2017) Norms of egality and the gap between attitudes and behaviour amongst Polish students: An experimental study. Economics & Sociology, 10(3), 220-236
- They used the extensive form of the ultimatum game. However, the proposer was asked the value of the minimum offer he/she would accept. There was a significant negative correlation between the amount demanded and the declared minimal value of an acceptable offer. In the context of this paper, individuals who demand a large amount but are prepared to accept a low offer are likely to be reflective. One interesting aspect of this research was that there is a small percentage of players who stated that they would not be prepared to accept the offers that they actually presented the respondent. Such players might be described as being "agressive" rather than "reflective". It should be noted that the threshold declared in the M-P & R paper should not be directly compared to the threshold declared in the paper under review (in the M-P & R paper this declaration was purely hypothetical, while in the paper under review this declaration has [or at least might have] practical consequences).
- One future direction might be to analyse the relation between the form of the game played and the reflectiveness of players.
- The level of English is generally good, but there are number of vocabulary/grammatical errors that should be corrected. For example,
- "the intuitive actions are to sanction a norm violation" -> sanctioning the violation of a norm is an intuitive action l.174 "belong to educations" -> are available l. 312 "act rational and self-interested"-> act rationally in accordance with their self-interest l. 343 this is a concern for the anonymity -> Hence, there is a concern regarding anonymity
- In conclusion, the article may be interesting to the general readership, but the results are not particularly enlightening to those working in behavioral economics.
Round 2
Reviewer 2 Report
My comments have been addressed and I feel that the article is now suitable for publication.